# Comparative Safety Analysis of Empagliflozin in Type 2 Diabetes Mellitus Patients with Chronic Kidney Disease versus Normal Kidney Function: A Nationwide Cohort Study in Korea

**DOI:** 10.3390/pharmaceutics15102394

**Published:** 2023-09-27

**Authors:** Ha Young Jang, In-Wha Kim, Jung Mi Oh

**Affiliations:** 1College of Pharmacy, Gachon University, Incheon 21936, Republic of Korea; hyjang@gachon.ac.kr; 2College of Pharmacy, Research Institute of Pharmaceutical Sciences, Seoul National University, Seoul 08826, Republic of Korea; iwkim2@hanmail.net

**Keywords:** empagliflozin, chronic kidney disease, diabetes, adverse effects, real-world evidence

## Abstract

Background: Empagliflozin has been shown to reduce cardiovascular morbidity and mortality in patients with type 2 diabetes. Various research on its efficacy in patients with chronic kidney disease (CKD) have been actively conducted. So far, few studies have investigated the safety of these adverse effects specifically in Asians with CKD. We aim to address these safety concerns on a patient population of Asian CKD patients using real-world data. Methods: We conducted a retrospective cohort study using health insurance data from the Korean Health Insurance Review & Assessment Service and compared safety outcomes between empagliflozin and sitagliptin in 26,347 CKD patients diagnosed with diabetes. Adverse outcomes, including major adverse cardiac events (MACEs), all-cause mortality, myocardial infarction (MI), stroke, and hospitalization for heart failure (HHF), among others, were assessed. Results: Among a 1:1 matched cohort (6170 on empagliflozin, 6170 on sitagliptin), empagliflozin was associated with a significant reduction in MACEs, all-cause mortality, MI, hospitalization for unstable angina, coronary revascularization, HHF, hypoglycemic events, and urinary tract infections, but increased the risk of genital tract infections. No significant changes were observed for transient ischemic attack, acute kidney injury, volume depletion, diabetic ketoacidosis, thromboembolic events, and fractures. Conclusions: The usage of empagliflozin in diabetic CKD patients shows a significant reduction in many adverse outcomes compared to sitagliptin, but with an increased risk of genital tract infections. These findings provide evidence for future clinical decision-making around the use of empagliflozin in Asian CKD patients.

## 1. Introduction

In recent years, various research has emerged investigating the role of empagliflozin in managing various aspects of diabetic complications. One of the studies that broadened the scope of empagliflozin usage was the EMPA-REG OUTCOME trial [1]. The trial demonstrated significant reductions in cardiovascular morbidity and mortality in patients with type 2 diabetes mellitus (T2DM) and established cardiovascular disease. This paved the way for a paradigm shift in the treatment strategy, with empagliflozin at the forefront of therapeutic regimens and now widely being recommended as a second-line diabetic medication after metformin [2,3]. In 2021, it was also approved for heart failure indications in patients without diabetes [4].

Chronic kidney disease (CKD) is prevalent in about 40% of individuals with type 2 diabetes, and patients with CKD are at a higher risk of adverse outcomes due to their compromised renal function [5]. Patients with an estimated glomerular filtration rate (eGFR) below 30 mL/min/1.73 m^2^ were not included, and over 70% of the patients in the trials had normal kidney function in the pivotal study. [1] To address these issues, several randomized controlled trials (RCTs) have been conducted to evaluate the various clinical efficacy and adverse effects of empagliflozin in patients with CKD. The EMPEROR-Reduced study demonstrated enhanced effectiveness in managing heart failure and deteriorating kidney function for the group with eGFRs of less than 60 mL/min/1.73 m^2^ [6]. The EMPA-REG OUTCOME study indicated there was no notable variance in side effects between the group with an eGFR less than 60 mL/min/1.73 m^2^ and those with standard kidney function [7]. Furthermore, the EMPA-KIDNEY study, which recently encompassed up to 30% of participants with an eGFR below 30 mL/min/1.73 m^2^, revealed that empagliflozin continues to be advantageous for declining kidney function and cardiovascular disease [8]. A pooled analysis of safety results from RCTs reported a beneficial effect on hyperkalemia and edema [9]. It seems the benefits of empagliflozin were generally consistent across a range of eGFR values [10].

Additionally, the real-world effectiveness of empagliflozin in patients with diabetes and CKD has recently been explored. Various studies investigated the long-term effects of empagliflozin on renal outcomes in a real-world setting and concluded that the drug did slow the progression of kidney disease in these patients [11,12]. Htoo et al. reported that empagliflozin significantly reduces the risk of MACEs (but not liraglutide) or HHF when compared to liraglutide or sitagliptin [13]. Furthermore, SGLT2 inhibitors have also been reported to reduce blood pressure, uric acid, and microalbuminuria or podocyturia, indicating they might have structural benefits for glomerular health [14]. 

However, most of the prior research has been conducted in the US or Europe. In fact, in the EMPA-REG OUTCOME trial, over 70% of participants were Caucasian, with merely 20% being Asian [1]. The effect or adverse drug reaction to a drug could vary based on genetic factors or medical practices across different racial groups. For instance, Asians are often reported to have a lower rate of cardiovascular disease risk compared to other ethnicities [15]. Differences in diabetes’ epidemiology and pathophysiology exist across various racial and ethnic groups [16,17,18]. Asian individuals with type 2 diabetes often receive their diagnosis relatively early, with about 20% being diagnosed before turning 40 years old [19]. This group with early onset diabetes also has an elevated risk for complications compared to those with a later onset [20]. Growing research indicates that Asians with type 2 diabetes have a heightened risk for CKD compared to other ethnicities [21,22].

A primary concern regarding adverse drug reactions to empagliflozin among CKD patients revolves around diabetic ketoacidosis (DKA) and dehydration [4]. So far, few studies have investigated the safety of these adverse effects specifically in Asians with CKD. While existing evidence does highlight beneficial renal outcomes and possible cardiovascular benefits, the risks associated with adverse reactions like diabetic ketoacidosis and dehydration need careful examination, especially for Asian populations. Although there have been numerous retrospective studies analyzing the effectiveness and safety of empagliflozin, none have compared results between CKD and non-CKD patients. Real-world evidence (RWE) studies hold value as they offer insights over longer durations and encompass a broader patient demographic [23,24]. They might be valuable in scenarios where RCT data may be lacking or unrepresentative of Asian population. Thus, our objective is to address these safety aspects through undertaking an RWE study focused on Korean CKD patients, utilizing insurance claim data.

## 2. Materials and Methods

### 2.1. Study Design and Sources

This was a retrospective cohort study evaluating the impact of empagliflozin compared to sitagliptin on safety outcomes in patients with CKD and T2DM. The analyzed health insurance data was officially provided by the Korean Health Insurance Review & Assessment Service (HIRA) [25]. In Korea, enrollment in the National Health Insurance (NHI) program is mandatory for 97% of the population. Healthcare facilities like clinics and hospitals file claims with the HIRA service to get reimbursement for both inpatient and outpatient care. These claims include details such as diagnoses coded according to the International Classification of Diseases, 10th revision (ICD10), procedures performed, prescription records, and demographic details. 

### 2.2. Ethical Approval

Given that all participants were anonymized using a randomized identification number, there was no requirement for written informed consent. The study received approval from the Institutional Review Board of Seoul National University (IRB No. E2101/001-003) and adhered to the Strengthening the Reporting of Observational Studies in Epidemiology (STROBE) guidelines [26]. 

### 2.3. Study Patients

This study included patients diagnosed with T2DM between 2016 and 2018. All included patients were adults (18 years or older) with established CKD (N18.4, N18.5, N18.6, E11.2, and E13.2) and T2DM (E11–E14) who were prescribed either empagliflozin or sitagliptin for the first time. We selected an active comparator (sitagliptin) as a proxy for the placebo. We chose an active comparator, sitagliptin, to stand in for the placebo due to its well-established use in observational studies. This decision was made because non-user comparator groups in such studies can show significant differences from actively treated patients, in contrast to RCTs [27]. Utilizing an active comparator can also aid in minimizing the risk of immortal time bias. Numerous other research has also used Dipeptidyl peptidase-4 (DPP4) inhibitors as comparators while evaluating the safety of SGLT-2, given the established safety records of DPP4 inhibitors [28,29,30,31,32]. The index date was determined to be the very first date each drug (empagliflozin or sitagliptin) was prescribed. The study period was before the new 2022 American Diabetes Association guidelines on type 2 diabetes were introduced [33], and empagliflozin and sitagliptin were commonly considered as second- or third-line options post metformin. In some cases, empagliflozin was prescribed after a DPP-4 inhibitor due to safety concerns associated with its use. 

### 2.4. Key Variables

A total of 16 adverse effects including major adverse cardiovascular events (MACEs), all-cause mortality, myocardial infarction [MI], hospitalization for unstable angina, coronary revascularization, stroke, transient ischemic attack (TIA), hospitalization for heart failure (HHF), hypoglycemic events, urinary tract infections (UTIs), genital tract infections (GTIs), volume depletion, acute kidney injury (AKI), DKA, thromboembolic events, and bone fracture were analyzed. In each cohort, individuals with a history of these adverse effects were excluded, and separate cohorts were constructed. The operational definitions of outcomes were determined using the Korean Standard Classification of Diseases-7 codes or procedure codes (Appendix A). To minimize the influence of potential confounding variables, such as selection bias, we included a total of 71 covariates. These covered demographics, comorbidities, and disease/outcome-specific variables. All these covariates were evaluated within the year preceding the index date.

### 2.5. Statistical Analysis

Statistical analyses were performed for the intention-to-treat population. Patients were followed until the earliest occurrence of any of the following events: an outcome event, the date of the last follow-up, the date of switching diabetic medication to the other comparison group, or the end of the study period. The maximum follow-up period was set at 48 months. Empagliflozin users were matched 1:1 with sitagliptin users, and the distribution of the propensity score was inspected [34]. A standardized difference greater than 0.1 was considered indicative of an imbalance [35]. The Cox proportional hazard regression model was used to estimate the sex- and age-adjusted hazard ratio (aHR) of empagliflozin for adverse outcomes, with a 95% confidence interval (CI).

Sensitivity analyses were performed in two ways. First, patients who received at least 1 dose of each study drug were observed until ≤30 d after a patient’s last intake of medication. Additionally, we followed up patients who received the study drug for ≥30 d (cumulative) including events that only occurred ≤30 d after a patient’s last intake of medication (‘as-treated’ analysis). Analyses were performed with SAS Enterprise Guide version 7.1 (SAS Institute Inc., Cary, NC, USA).

### 2.6. Comparative Analysis

Using the same study design, patients in the normal kidney function (NKF) group were identified just like the CKD patients. The comparative analysis was conducted in a comparable fashion on patients with NKF and contrasted with the results obtained from patients with CKD. They were evaluated for the same set of 16 outcomes including MACEs, all-cause death, MI, and more. The statistical analysis, involving Cox proportional hazard regression and propensity score matching, was performed identically to maintain consistency. Thus, the outcomes in NKF patients were compared to the outcomes in CKD patients, thereby providing a comparison between the two distinct patient groups.

## 3. Results

### 3.1. Demographics

A total of 932,465 patients diagnosed with type 2 diabetes and treated with either empagliflozin or sitagliptin were identified. From this group, 384,579 new users of these medications remained (Figure 1). Patients diagnosed with CKD were then selected, resulting in an eligible study cohort of 26,347 patients (6211 on empagliflozin, 20,136 on sitagliptin). The data revealed that sitagliptin users were older and had more frequent clinic visits (both inpatient and outpatient) compared to empagliflozin users (Table 1 and Appendix A). Moreover, a higher prevalence of coronary artery disease, including stroke, was observed among sitagliptin users relative to empagliflozin users.

A successful match was achieved between 6170 empagliflozin users and sitagliptin users. After matching, the differences in age, frequency of clinic visits, index date, cardiovascular risk factors, comedications, and comorbidities between the two groups were considerably diminished, resulting in a well-balanced cohort. All 71 covariates showed standardized differences well below 0.1. The median follow-up period was recorded as 0.9 years, with the median duration of prescription for anti-diabetic medications during the follow-up period being 0.9 years (interquartile range 0.2–2.3 years). The mean age of the patients was noted as 50.9 years, with men constituting 56.7% of the cohort (*n* = 6998). The baseline characteristics of patients with normal kidney function are presented in Appendix A.

### 3.2. Risk of Safety Outcomes 

The use of empagliflozin was associated with a notable reduction in the risk of MACEs, with an aHR of 0.74 (95% CI: 0.64–0.85) (Table 2). The risk of all-cause mortality was also significantly reduced when treated with empagliflozin (aHR 0.47, 95% CI: 0.33–0.68). In the context of MI, empagliflozin usage resulted in a lower risk, as denoted by an aHR of 0.60 (95% CI: 0.45–0.81). However, for stroke, it did not show a significant difference in risk, with an aHR of 0.92 (95% CI: 0.74–1.13). Patients receiving empagliflozin demonstrated a decreased risk of hospitalization for unstable angina, presenting an aHR of 0.67 (95% CI: 0.57–0.79). Likewise, the risk of coronary revascularization was marginally reduced with an aHR of 0.79 (95% CI: 0.65–0.97). HHF showed a reduction in risk with an aHR of 0.66 (95% CI: 0.55–0.80). Hypoglycemic adverse events also had a slightly lower risk with empagliflozin treatment, as indicated by an aHR of 0.78 (95% CI: 0.62–0.97). Additionally, the risk of UTIs was slightly reduced, with an aHR of 0.90 (95% CI: 0.82–0.98). Notably, the risk of GTIs was increased with empagliflozin, with an aHR of 1.43 (95% CI: 1.27–1.61). There were no significant changes in the risk of transient ischemic attack, AKI, volume depletion, DKA, thromboembolic events, and fractures as indicated by the aHRs close to 1.

### 3.3. Sensitivity Analysis

For sensitivity analysis, after the follow-up of patients who received at least one dose of study drugs until ≤30 d after the last intake of medication, similar results were obtained in all 16 outcomes. Additional sensitivity analysis (including patients who received study drugs for ≥30 d including only events that occurred ≤30 d after a patient’s last intake of medications) did not produce meaningful changes in the study findings (Appendix A).

### 3.4. Comparative Analysis: CKD Group versus NKF Group

In a comparative analysis, we observed that CKD patients who used empagliflozin showed a protective effect against cardiovascular diseases. These included associations with MACEs, MI, coronary revascularization, angina, HHF, hypoglycemia, UTIs, GTIs, volume depletion, and thromboembolic events, as depicted in Figure 2. A notable observation was the risk of death in the CKD group with an aHR of 0.47 (95% CI: 0.33–0.68). Empagliflozin in the CKD group was not significantly associated with stroke, with an aHR of 0.92 (95% CI: 0.74–1.13); AKI, with an aHR of 0.94 (95% CI: 0.76–1.17); DKA, with an aHR of 0.96 (95% CI: 0.60–1.54); or fractures, with an aHR of 1.03 (95% CI: 0.92–1.14).

For NKF patients, the HRs for empagliflozin use were generally similar between both groups, except some outcomes. The risk of stroke was noted with an aHR of 0.74 (95% CI: 0.69–0.79). The association with the risk of AKI was marked by an aHR of 0.75 (95% CI: 0.69–0.80). Additionally, the risk of fractures was noted with an aHR of 0.91 (95% CI: 0.88–0.94) in the NKF group.

## 4. Discussion

This RWE study explored the safety outcomes of empagliflozin use in T2DM patients with CKD. To our knowledge, this is the first RWE study to examine a range of safety outcomes in Asian CKD patients. Despite concerns that these patients might not respond well to empagliflozin due to the reduced glucose reabsorption capabilities of their kidneys [4], the study found a substantial decrease in cardiovascular event risks in Asian CKD patients. These results are consistent with findings from the EMPA-REG OUTCOME, EMPEROR-REDUCED, and EMPA-KIDNEY studies [6,7,8]. 

The kidneys play a crucial role in the underlying mechanisms of T2DM. Firstly, in healthy individuals, the kidneys account for about 20% to 25% of the body’s endogenous glucose production during fasting, due to the process of gluconeogenesis. Secondly, the kidneys are pivotal in managing blood glucose levels as they oversee both glucose filtration and reabsorption processes [36,37]. At this point, empagliflozin has received more attention for its glucosuria mechanism. Moreover, empagliflozin’s benefits in CKD patients appear to extend beyond just inducing glucosuria, positively impacting cardiovascular and renal dynamics, reducing inflammation, and providing protection to end-organs. Several mechanisms have been proposed for the effects of empagliflozin, including its role in natriuresis, ketogenesis, lipid metabolism, improvements in cellular and endothelial functions, and antiproliferative effects on certain types of cancer. One of the mechanisms involves aiding sodium excretion [38], which in turn activates transforming growth factor, offering hemodynamic protection to the kidneys [39,40,41]. Concerns have been primarily raised regarding AKI when empagliflozin is used in patients with CKD [42] due to vasoconstriction or volume depletion; however, such events are usually temporary. Our study did not identify any enhanced risks of AKI or volume depletion in Asian CKD patients. Additionally, empagliflozin boosts endogenous G production that promotes lipid oxidation and ketone body utilization. Studies have reported an increase in lipolysis and a reduction in visceral fat [43]. In CKD patients with decreased renal function, the development of DKA has been a concern due to decreased ketone excretion; however, this study found that empagliflozin did not increase the risk of DKA. Furthermore, empagliflozin has been associated with improved arterial vascular stiffness and reduced resistance [44,45]. It has shown a decline in both cardiac and vascular sympathetic nerve activities, and it exhibits anti-inflammatory properties [46,47] and mitigates fibrosis [45,48]. All these factors provide a strong rationale for its potential cardiovascular benefits, especially in CKD patients with deteriorated renal function. However, our study did not find significant changes in the risk for stroke, transient ischemic attack, thromboembolic events, and fractures.

Our research strength lies in its comprehensive analysis of various adverse effects. To date, only a few studies have investigated the effectiveness or safety of empagliflozin in Asian patients with CKD. Sugiyama et al. reported that SGLT2 inhibitor enhanced kidney protection, leading to a marked improvement in the reduction in eGFR and proteinuria in Japanese patients [12]. Another study also showed a reduction in risk for developing end-stage kidney disease in Singaporean patients [49]. However, a significant limitation of these studies is their focus solely on CKD’s progression, neglecting to consider cardiovascular events and side effects. We examined a range of safety outcomes in Asian CKD patients and concluded that the pattern of results closely aligns with those observed in Western populations. A few key points should be considered when interpreting these results. For cardiovascular outcomes, Asian individuals often exhibit a stronger reaction to antihypertensive medications that influence the renin–angiotensin–aldosterone system. [50]. Another study also indicates racial distinctions in plasma renin activity levels [51]. This suggests reducing overall sodium content as a potential therapeutic target, showing why Asian individuals might be also responsive to empagliflozin. It is reported that GTIs were considerably less frequent in Asians than in Western populations, and this could be attributed to better hygiene practices in Asia [52]. However, GTIs also seem to be an issue with the use of empagliflozin in the Asian population. 

For those with significant renal impairment in CKD (<30 mL/min/1.73 m^2^), the area under the concentration–time curve of empagliflozin increased 1.7–2.7-fold, and its half-life was increased by 1.4-fold compared to NKF patients [53,54]. Indeed, our comparative analyses have shown that Asian CKD patients showed a higher occurrence of adverse reactions irrespective of being on sitagliptin or empagliflozin compared to NKF patients. Nevertheless, the HRs for empagliflozin use in relation to various outcomes were generally consistent between both groups, with a few exceptions. SGLT2 inhibitors are as effective to use in CKD patients as they are in NKF patients while preventing a variety of cardiovascular events. The cardiovascular safety benefits of empagliflozin appear to be further maximized in CKD patients. However, the comparative analysis did reveal no significant results for stroke, AKI, and fractures in CKD patients compared to NKF patients. Other RWE studies have also reported safety in other NKF patients with these results (stroke [55], AKI [56,57,58,59], and fractures [60]). This could be explained by the underlying disease state and co-morbidities in CKD patients and warrants further research to explore these risks in more detail. Even though empagliflozin did not reveal any new safety concerns, these findings underline the significance of tailored therapeutic strategies in CKD patients. This fact emphasizes the importance of patient education and close monitoring to prevent and manage such complications.

This study’s results need to be considered within the limitations of an observational study. Our research employs a retrospective cohort methodology, and the HIRA database does not incorporate all details, such as lab outcomes for blood glucose tests, urine culture tests, or body weight. Despite our effort to control all potential confounders, there still may be residual confounding factors present. It should also be noted that the control utilized in this research was sitagliptin, not a genuine placebo. This could potentially impact the study’s external validity. Furthermore, the follow-up period in this study was relatively short, potentially limiting the ability to detect the long-term effects of the medications.

## 5. Conclusions

In conclusion, our study results suggest reassuring the efficacy and safety profiles of empagliflozin in CKD patients, and further supports its use in Asian population. As the number of patients with CKD continues to grow globally, the real-world evidence in this study could serve as a key step in providing evidence for future clinical decision-making around the use of empagliflozin in Asian CKD patients.

## Figures and Tables

**Figure 1 pharmaceutics-15-02394-f001:**
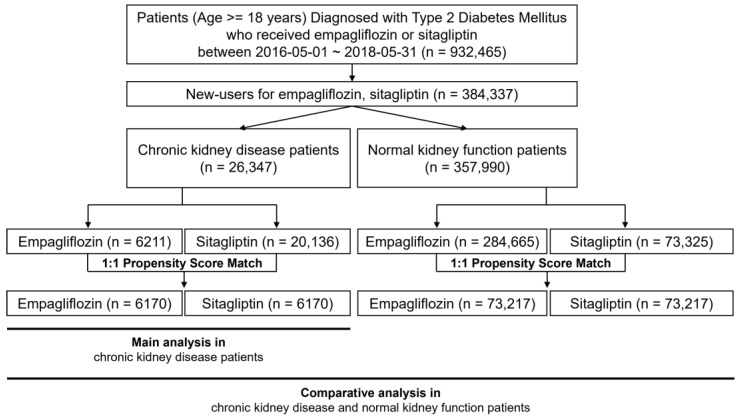
Study flow chart.

**Figure 2 pharmaceutics-15-02394-f002:**
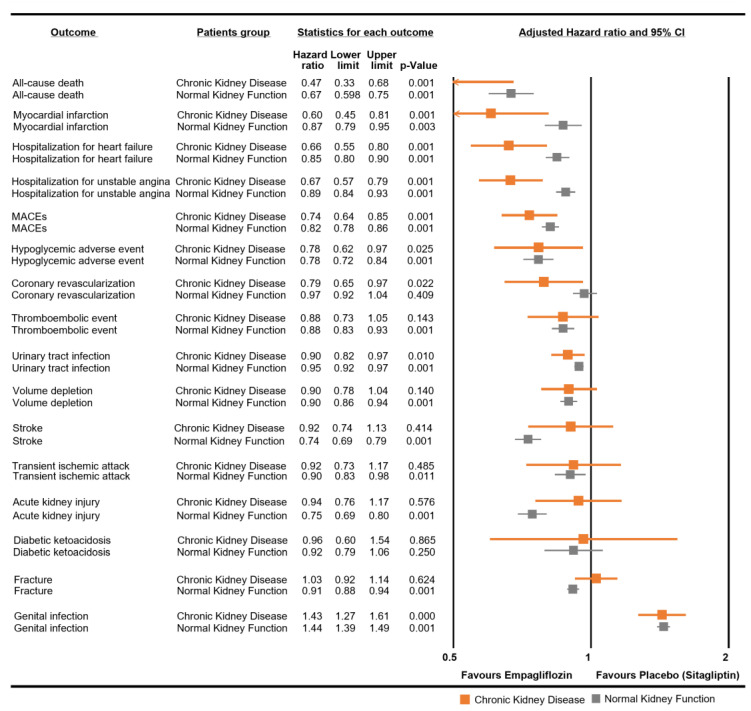
Comparative analysis of empagliflozin for each safety outcome in chronic kidney disease and normal kidney function patients. Normal kidney function patients refer to those not diagnosed with chronic kidney disease.

**Table 1 pharmaceutics-15-02394-t001:** Baseline characteristics of matched cohort.

Variables	Sitagliptin n = 6170	Empagliflozin n = 6170	STD
Sex, male	3534 (57.3)	3464 (56.1)	−0.02
Age, year	50.8 ± 12.8	50.9 ± 12.1	0.01
Normal	5842 (94.7)	5846 (94.8)	0.004
Medicaid	307 (5)	304 (4.9)	
No charge	21 (0.3)	20 (0.3)	
Number of inpatient visits	0.5 ± 1.3	0.5 ± 1.3	0.003
Number of outpatient visits	29.0 ± 28.4	29.0 ± 28.4	0.002
Index year			
2016	1545 (25)	1564 (25.4)	0.02
2017	3106 (50.3)	3057 (49.6)	
2018	1519 (24.6)	1549 (25.1)	
Charlson comorbidity index			
0	2 (0)	1 (0)	0.02
1	416 (6.7)	408 (6.6)	
2	503 (8.2)	523 (8.5)	
3	5249 (85.1)	5238 (84.9)	
CV risk factor			
CAD	1946 (31.5)	2054 (33.3)	0.04
Multi vessel CAD	1080 (17.5)	1126 (18.3)	0.02
MI	110 (1.8)	95 (1.5)	−0.02
CABG	444 (7.2)	430 (7)	−0.01
Stroke	190 (3.1)	181 (2.9)	−0.01
PAD	170 (2.8)	148 (2.4)	−0.02
DM circulation	1078 (17.5)	1083 (17.6)	0.002
DM foot	0 (0.0)	0 (0.0)	0
DM nephropathy	5789 (93.8)	5828 (94.5)	0.03
DM neuropathy	1111 (18)	1089 (17.7)	−0.01
DM other complications	4101 (66.5)	4085 (66.2)	−0.01
Hyperglycemia	126 (2)	140 (2.3)	0.02
CV risk factor			
Hypertension	4312 (69.9)	4330 (70.2)	0.006
Edema	704 (11.4)	696 (11.3)	−0.004
Kidney stone	120 (1.9)	123 (2.0)	0.004
Osteoarthritis	1832 (29.7)	1808 (29.3)	−0.01
Other arthritis	1672 (27.1)	1698 (27.5)	0.01
PUD	1619 (26.2)	1596 (25.9)	−0.01
Pancreatitis	126 (2)	124 (2)	−0.002
UC	12 (0.2)	9 (0.2)	−0.01
Crohn	3 (0.1)	4 (0.1)	0.000
Asthma	946 (15.3)	899 (14.6)	−0.02
COPD	182 (3)	164 (2.7)	−0.02
Bladder stone	2 (0)	3 (0.1)	0.01
Dementia	516 (8.4)	525 (8.5)	0.005
Electrolyte imbalance	576 (9.3)	617 (10)	0.02
Glaucoma/cataract	1775 (28.8)	1766 (28.6)	−0.003
HONK	40 (0.7)	44 (0.7)	0.01
HTN nephropathy	354 (5.7)	377 (6.1)	0.02
Hyperthyroid disease	170 (2.8)	175 (2.8)	0.005
Hypothyroid disease	722 (11.7)	685 (11.1)	−0.02
Osteomyelitis	32 (0.5)	38 (0.6)	0.01
Pneumonia	493 (8)	457 (7.4)	−0.02
Skin infection	296 (4.8)	290 (4.7)	−0.005
Metformin	4731 (76.7)	4752 (77)	0.008
Insulins	1575 (25.5)	1620 (26.3)	0.02
SUs	3154 (51.1)	3178 (51.5)	0.01
Glitazones	978 (15.9)	954 (15.5)	−0.01
GLP-1 agonists	65 (1.1)	78 (1.3)	0.02
AGIs	225 (3.7)	195 (3.2)	−0.03
Meglitinides	98 (1.6)	113 (1.8)	0.02
Anticoagulants	2837 (46)	2825 (45.8)	0.01
Antiplatelets	2783 (45.1)	2768 (44.9)	−0.05
Heparins	190 (3.1)	182 (3)	−0.01
Thrombolytics	3 (0.1)	3 (0.1)	0.01
Statins	4750 (77)	4775 (77.4)	0.01
Other lipid lowerings	1129 (18.3)	1165 (18.9)	0.01
Nitrates	451 (7.3)	443 (7.2)	−0.005
Digoxin	392 (6.4)	391 (6.3)	−0.001
ACEIs	255 (4.1)	252 (4.1)	−0.002
ARBs	3752 (60.8)	3723 (60.3)	−0.01
Entresto	0 (0)	2 (0)	0.03
Other anti-HTNs	3009 (48.8)	2995 (48.5)	−0.005
Loop diuretics	638 (10.3)	632 (10.2)	−0.003
Other diuretics	1509 (24.5)	1469 (23.8)	−0.02
Antianxieties	2131 (34.5)	2176 (35.3)	0.02
Antipsychotics	163 (2.6)	168 (2.7)	0.005
Antidepressants	891 (14.4)	893 (14.5)	0.001
Dementia	516 (8.4)	525 (8.5)	0.005
Antiparkinsons	114 (1.9)	117 (1.9)	0.004
Anticonvulsants	90 (1.5)	90 (1.5)	0
NSAIDs	4828 (78.3)	4779 (77.5)	−0.02
Bisphosphonates	140 (2.3)	156 (2.5)	0.02
Opioids	2610 (42.3)	2639 (42.8)	0.01

Values are represented as mean ± standard deviation or number (%); ACEis, angiotensin-converting enzyme inhibitors; AGIs, α-glucosidase inhibitors; ARBs, angiotensin II receptor blockers; CABG, coronary artery bypass graft; CAD, coronary artery disease; COPD, chronic obstructive pulmonary disease; CV, cardiovascular; DM, diabetes mellitus; HONK, hyperglycaemic hyperosmolar nonketotic coma; HTN, hypertensive; MI, myocardial infarction; NSAIDs, non-steroidal anti-inflammatory drugs; PAD, peripheral artery disease; PUD, peptic ulcer disease; STD, standardized difference; SUs, sulfonylureas; UC, ulcerative colitis.

**Table 2 pharmaceutics-15-02394-t002:** Hazard ratios of empagliflozin for each safety outcome.

	Events	Person-Year	Hazard Ratio (95% CI)	
			Unadjusted	Adjusted
MACEs				
Sitagliptin	470	16,824		
Empagliflozin	342	16,945	0.72 (0.63, 0.83) *	0.74 (0.64, 0.85) *
All-cause death				
Sitagliptin	96	12,391		
Empagliflozin	43	12,814	0.43 (0.30, 0.62) *	0.47 (0.33, 0.68) *
Myocardial infarction				
Sitagliptin	120	17,104		
Empagliflozin	74	17,320	0.61 (0.46, 0.82) *	0.60 (0.45, 0.81) *
Stroke				
Sitagliptin	179	16,568		
Empagliflozin	161	16,765	0.89 (0.71, 1.10)	0.92 (0.74, 1.13)
Hospitalization for unstable angina				
Sitagliptin	364	15,428		
Empagliflozin	254	15,829	0.68 (0.58, 0.80) *	0.67 (0.57, 0.79) *
Coronary revascularization				
Sitagliptin	220	15,670		
Empagliflozin	178	15,851	0.80 (0.65, 0.98) *	0.79 (0.65, 0.97) *
Transient ischemic attack				
Sitagliptin	143	16,422		
Empagliflozin	131	16,552	0.91 (0.72, 1.15)	0.92 (0.72, 1.17)
Hospitalization for heart failure				
Sitagliptin	265	16,334		
Empagliflozin	175	16,640	0.65 (0.54, 0.79) *	0.66 (0.55, 0.80) *
Hypoglycemic adverse event				
Sitagliptin	177	16,431		
Empagliflozin	138	16,666	0.77 (0.62, 0.96) *	0.78 (0.62, 0.97) *
Urinary tract infections				
Sitagliptin	1112	9909		
Empagliflozin	1032	10,231	0.90 (0.83, 0.98) *	0.90 (0.82, 0.98) *
Genital infections				
Sitagliptin	459	12,300		
Empagliflozin	639	12,025	1.42 (1.26, 1.60) *	1.43 (1.27, 1.61) *
Acute kidney injury				
Sitagliptin	168	16,402		
Empagliflozin	159	16,647	0.94 (0.75, 1.16)	0.94 (0.76, 1.17)
Volume depletion				
Sitagliptin	413	15,336		
Empagliflozin	375	15,436	0.90 (0.78, 1.04)	0.90 (0.78, 1.04)
Diabetic ketoacidosis				
Sitagliptin	35	17,089		
Empagliflozin	34	17,221	0.96 (0.60, 1.54)	0.96 (0.60, 1.54)
Thromboembolic event				
Sitagliptin	257	16,042		
Empagliflozin	226	16,227	0.87 (0.73, 1.04)	0.88 (0.73, 1.05)
Fracture				
Sitagliptin	667	13,200		
Empagliflozin	684	13,337	1.02 (0.91, 1.13)	1.03 (0.92, 1.14)

* Statistically significant; hazard ratio was adjusted for age and sex.

## Data Availability

This study used HIRA research data (M20200708641) from HIRA. The data that support the findings of this study are available from HIRA, but restrictions apply to the availability of these data, which were used under license for the current study, and so are not publicly available. Data are, however, available from the authors upon reasonable request and with the permission of HIRA.

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
