# Peer review of "Comparative Safety Analysis of Empagliflozin in Type 2 Diabetes Mellitus Patients with Chronic Kidney Disease versus Normal Kidney Function: A Nationwide Cohort Study in Korea"

_pharmaceutics, 2023, doi:10.3390/pharmaceutics15102394_

Round 1
Reviewer 1 Report
Author presented a study aim to address these safety concerns on the type 2 diabetes patient population with CKD. They conducted a retrospective cohort study to analyze the comparing safety outcomes between empagliflozin and sitagliptin in CKD patients with diabetes. They concluded that empagliflozin administration in diabetic CKD patients presented a significant reduction in many adverse outcomes compared to sitagliptin, such as MACEs, all-cause mortality, MI, and hospitalization for unstable angina. Their findings provide evidence for therapeutic options for the usage of empagliflozin in CKD patients.
Comments to authors:
1. It is needed to discuss the possible mechanisms of the clinical correlation of the diabetics and CKD.
2. There should be some discussion about the possible mechanisms of the benefits of empagliflozin and sitagliptin treatment on the kidney functions of the diabetes patients.
3. The authors should update more currently studies that also recognized the evidence on the comparative effectiveness of empagliflozin vs alternative second-line glucose-lowering agents in T2D patients. Such as “Comparative Effectiveness of Empagliflozin vs Liraglutide or Sitagliptin in Older Adults With Diverse Patient Characteristics. JAMA Netw Open. 2022 Oct 3;5(10):e2237606. doi: 10.1001/jamanetworkopen.2022.37606.” Please discuss it. It is needed to point out the novelty and advantages of their study.
4. The authors should update more currently in vivo or in vitro studies that also recognized the benefits of empagliflozin on the treatment of CKD or diabetes.
Author Response
Dear Editor and Reviewer,
We were pleased to have an opportunity to revise our manuscript now entitled “Comparative Safety Analysis of Empagliflozin in Type 2 Diabetes Mellitus Patients with Chronic Kidney Disease versus Normal Kidney Function: A Nationwide Cohort Study in Korea". In the revised manuscript, we have carefully considered reviewers’ comments and suggestions. As instructed, we have attempted to succinctly explain changes made in reaction to all comments. We reply to each comment in point-by-point fashion. We have color coded revised manuscript as text. The responses to the concerns raised by reviewers are below and are color coded as follows: a) Comments from editors or reviewers are shown as text; b) Our responses are shown as text, table, or figure.
The reviewers’ comments were constructive overall, and we are appreciative of such constructive feedback on our original submission. After addressing the issues raised, we feel the quality of the paper is much improved.
Please see the attachment.

Reviewer 2 Report
Comments to the author(s)
In this retrospective cohort study by the authors using data from Korean Health Insurance Review & Assessment Service (HIRA), they aimed to address safety concerns of SGLT2 inhibitors by conducting a real-life study on a patient population with CKD. The manuscript is written well, the text is fluent and easy to follow and, the statistical analyzes are very satisfying.
However, in this study; there are several minor points that can be revised.
1) Page 1, lines 39-40: ’’However, the applicability of these trials' findings to the chronic kidney disease (CKD) population is still under uncertainty.’’ and, page 10, lines 216-219: ’’To our knowledge, this is the first study to examine a range of safety outcomes in a patient population comprised exclusively of CKD patients. Notably, our study further extends
these findings to a population with CKD, a population that was underrepresented in the EMPA-REG OUTCOME trial.’’ These statements are not valid according to current literature.
Please refer to the most recent study:
’’The EMPA-KIDNEY Collaborative Group. Empagliflozin in Patients with Chronic Kidney Disease. N Engl J Med. 2023 Jan 12;388(2):117-127. doi:
10.1056/NEJMoa2204233.’’
’’EMPA-KIDNEY ClinicalTrials.gov number, NCT03594110’’
In the EMPA-KIDNEY clinical trial and its resulting published paper, approximately 6600 cases of CKD with eGFR 20-90 were randomized. The Safety Outcomes and Adverse Events were also explicitly addressed.
2) In addition, in a recent study presenting real-life data, Durcan et al. prospectively investigated the effect of SGLT2 inhibitors in reducing podocyturia in patients with CKD.
This study is also recommended to be referred.
Please refer to the most recent study:
’’Durcan E. et al. Effects of SGLT2 inhibitors on patients with diabetic kidney disease:A preliminary study on the basis of podocyturia. J Diabetes. doi: 10.1111/1753-0407.13261. PMID: 35229458’’
3) Page 10, lines 221-223: ’’Our study further augments these results, demonstrating that these benefits extend to patients with eGFR below 30 mL/min/1.73 m2, a population that was not included in the meta-analysis.’’ This sentence cannot be presented as a true proposition. Because, it was presented as a difference from the literature that patients with eGFR <30 mL/min/1.73 m2 were included in the study, but no GFR data were available when the entire study was examined. The existence of GFR data should be explained or the issue should not be discussed in terms of GFR.
Author Response

(The authors gave the same response as above.)

Reviewer 3 Report
Dear author (s)
Your research manuscript entitled "Safety Outcomes of Empagliflozin in Type 2 Diabetes Mellitus 2 Patients with Chronic Kidney Disease: National-wide Cohort 3 Study", in this manuscript author investigate the use of certain medicine for the treatment of DM with CKD patient. The author has done well and also written scientifically well.
I have a few queries
1. Regarding Figure 2. the author has given a table in the form of an image.
2. In this image legend section, the author has written normal patient but can again explain in the legend section so that the reader can understand much better way.
Regards
Author Response

(The authors gave the same response as above.)

Reviewer 4 Report
Jang and colleagues submitted a manuscript entitled “Safety Outcomes of Empagliflozin in Type 2 Diabetes Mellitus Patients with Chronic Kidney Disease: National-wide Cohort Study” for publication in Pharmaceuticals. The manuscript reports on safety outcomes in the context of SGLT2-inhibitor use, in comparison to gliptin. In agreement with all previously published data, the authors find a significant benefit of SGLT2 inhibitors.
Unfortunately, the study reports nothing new, the benefits of SGLT2 inhibitors have been described in multiple studies before. As such, it seems the study does not add anything to current knowledge. This assumption is further supported by the fact that the authors did not present any of the findings as novel.
The authors do not seem to be aware of the current literature. E.g. the statement in the abstract “Its efficacy in patients with chronic kidney disease (CKD) and diabetes is unclear, as previous studies mostly included patients with normal kidney function.” is just wrong, multiple studies have shown efficacy in CKD, also in the absence of diabetes. Additional incorrect statements can be found throughout the paper, indicating a lack of knowledge on the subject. I strongly suggest the authors familiarize themselves with the current literature on SGLT2 inhibitors, and the rewrite their manuscript and present the findings in the context of the current knowledge.
I was hoping to see some data on malignancies, as there is a concern that SGLT2 inhibitors may impact immune surveillance, while at the same time potential benefits of SGLT2 inhibitors in oncology are being discussed. Unfortunately, it seems the authors have not investigated this issue at all, although data should be available, and this may add some novelty to the paper.
Overall, the study appears of very limited value and the text contains some inaccuracies. However, a thoroughly revised version based on a comprehensive search of the literature and including data on malignancy but may be of potential interest.
English language is fine.
Author Response

(The authors gave the same response as above.)

Reviewer 5 Report
Jang and colleagues investigated the efficacy of empagliflozin in diabetic patients with chronic kidney disease (CKD). Their analysis compared the safety effects of empagliflozin and sitagliptin using matched populations (6170 receiving empagliflozin, 6170 receiving sitagliptin). Outcomes were major adverse cardiac events (MACE), all-cause mortality, myocardial infarction (MI), stroke and hospitalisation for heart failure (HF). Empagliflozin significantly reduced MACE, all-cause mortality, MI, hospitalisation for unstable angina, coronary revascularisation, HHF, hypoglycaemic events and urinary tract infections, but increased the risk of sexually transmitted infections. No significant changes were observed for transient ischaemic attack, acute kidney injury, volume depletion, diabetic ketoacidosis, thromboembolic events and fractures. The results highlight the beneficial effect of empagliflozin compared to sitagliptin and draw attention to the increased risk of genital tract infections.
Comments and suggestions:
1. It is not mentioned in the title of the manuscript that the present study is a comparative analysis. I suggest a mention of a comparison with sitagliptin treatment in the title. I also recommend that the origin of the study population (Korea) be included in the title.
2. The Introduction is quite short and incomplete. I suggest a brief description of the specificities of Korea about diabetes and CKD.
3. The Introduction lacks justification, literature, and context for sitagliptin as a reference drug. I recommend that the introduction be completed. Why has this drug become the reference?
4. What criteria were used to diagnose diabetes in the sample population of this study? Also, in which cases does the doctor prescribe sitagliptin or empagliflozin?
5. The authors have previously published a similar study, which is referenced but not compared to the present one. I suggest that they highlight in the Introduction or Discussion section how the present study differs from those they have previously published (ref.: Jang HY, Kim IW, Oh JM. Using real-world data for supporting regulatory decision making: Comparison of cardiovascular and safety outcomes of an empagliflozin randomized clinical trial versus real-world data. Front Pharmacol. 2022 Aug 30;13:928121. doi: 10.3389/fphar.2022.928121. PMID: 36110539; PMCID: PMC9468970.)
6. The sub-section entitled "3.4. Comparative Analysis: CKD group versus NKF group" may be misleading. If I understand correctly, it is not the CKD and NKF groups that are compared, but the risk rate of those taking empagliflozin measured in aHR. Since there was no 1:1 matching for the CKD and NKF groups, the results presented are not comparable without correction for all possible covariates. Furthermore, no clear conclusions can be drawn between the analyses carried out on the CKD and NKF groups separately, as no statistics have been produced. For example, the statement that CKD patients have a higher risk of stroke than NKF patients is speculation without statistical analysis. I recommend that these results be confirmed by statistical analysis, or not presented in the manuscript.
7. I suggest that the Discussion be supplemented by a comparison with articles published on this topic and not discussed in the manuscript:
- Colbert GB, Madariaga HM, Gaddy A, Elrggal ME, Lerma EV. Empagliflozin in Adults with Chronic Kidney Disease (CKD): Current Evidence and Place in Therapy. Ther Clin Risk Manag. 2023 Feb 2;19:133-142. doi: 10.2147/TCRM.S398163. PMID: 36756278; PMCID: PMC9901477.
- Htoo PT, Tesfaye H, Schneeweiss S, et al. Comparative Effectiveness of Empagliflozin vs Liraglutide or Sitagliptin in Older Adults With Diverse Patient Characteristics. JAMA Netw Open. 2022;5(10):e2237606. doi:10.1001/jamanetworkopen.2022.37606
- Katherine R. Tuttle, Adeera Levin, Masaomi Nangaku, Takashi Kadowaki, Rajiv Agarwal, Sibylle J. Hauske, Amelie Elsäßer, Ivana Ritter, Dominik Steubl, Christoph Wanner, David C. Wheeler; Safety of Empagliflozin in Patients With Type 2 Diabetes and Chronic Kidney Disease: Pooled Analysis of Placebo-Controlled Clinical Trials. Diabetes Care 2 June 2022; 45 (6): 1445–1452. https://doi.org/10.2337/dc21-2034
8. In the Discussion of the results as well as in the conclusion, the authors make it clear that the results of the present study are not generalisable, but refer to the comparison of two specific drug treatments (empagliflozin vs. sitagliptin).
Minor comments:
- In the subsection Study Design and Sources, it is unnecessary to include the statement of the institutional review board, as this is a separate subsection.
- Table 1 is too large (2 and a half pages) and is not informative for the reader because of the matching of populations. I recommend that it be included as a supplement.
- For Table 2, please indicate which confounders have been adjusted for in the analyses.
Author Response

(The authors gave the same response as above.)

Reviewer 6 Report
The article “Safety Outcomes of Empagliflozin in Type 2 Diabetes Mellitus Patients with Chronic Kidney Disease: National-wide Cohort Study” has been submitted for review.
In recent years, various studies have emerged examining the role of empagliflozin in the treatment of various aspects of diabetic complications. Empagliflozin has been at the forefront of therapeutic options as it is now widely recommended as a second-line treatment for diabetes after metformin. In conclusion, the use of empagliflozin in patients with diabetic CKD showed a significant reduction in many adverse outcomes compared with sitagliptin, but with an increased risk of genital tract infections. These results provide evidence for future regulatory decisions regarding the use of empagliflozin in patients with CKD. The main side effects of concern in patients with CKD are diabetic ketoacidosis and dehydration, and concerns have been raised about the potential for additional damage to kidney function due to decreased drug elimination as kidney function declines. Therefore, the purpose of the work was to clarify the issue of the safety of this drug. This was a retrospective cohort study evaluating the effect of empagliflozin versus sitagliptin on safety outcomes in patients with CKD and type 2 diabetes mellitus. This study used insurance data that included demographic information, diagnoses, procedures, and patient prescription data. Adequate methods of statistical data analysis were used. The article is illustrated with 2 figures and 2 tables.
Conclusion. The article is of significant importance for practical healthcare and is interestingly written. However, it is advisable to add references to the literature section. It is also necessary to display the presence of statistically significant differences in all tables in the article.
Author Response

(The authors gave the same response as above.)

Round 2
Reviewer 4 Report
Thank you for addressing all my comments. It is unfortunate that the impact on oncological disease cannot be assessed, but I understand and accept the justification.
Slight improvement in the language may be helpful.
Reviewer 5 Report
I accept the authors' answers to my questions and comments.